# OPTION BOOSTING

## ABSTRACT

We introduce a novel approach to enhance stability and knowledge transfer in multi-task hierarchical reinforcement learning, specifically within the options framework. Modern Hierarchical Reinforcement Learning (HRL) algorithms can be prone to instability, due to the multilevel nature of the optimization process. To improve stability, we draw inspiration from boosting methods in supervised learning and propose a method which progressively introduces new options, while older options are kept fixed. In order to encourage generalization, each option policy has limited expressiveness. In order to improve knowledge transfer, we introduce the *Option Library*, a mechanism to share options across a population of agents. Our approach improves learning stability and allows agents to leverage knowledge from simple tasks in order to explore and perform more complex tasks. We evaluate our algorithm in MiniGrid and CraftingWorld, two pixel-based 2D grid-world environments designed for goal-oriented tasks, which allows compositional solutions.

## 1 INTRODUCTION

Hierarchical Reinforcement Learning (HRL) is an intuitive approach to building solutions to complex RL problems by composing simpler policies Wulfmeier et al. (2021); Vezhnevets et al. (2017); Sutton et al. (1999). The options framework Precup (2000); Sutton et al. (1999) has emerged as the main way for expressing HRL solutions to problems. This framework allows agents to employ temporally extended actions characterized by an initiation set, option-policy, and termination functions. Much work has been devoted to learning automatically a good set of options, either by learning subgoals Şimşek et al. (2005); McGovern and Barto (2001); Menache et al. (2002), learning end-to-end while solving a problem Bacon et al. (2017); Harb et al. (2018b); Zhang and Whiteson (2019), unsupervised learning Pathak et al. (2019); Eysenbach et al. (2018), or even random search Mann et al. (2015).

However, the process of training option sets is prone to instability, especially when used with deep neural networks. This is due to the fact that the optimization is layered, working on both the elements of the options and the way of choosing among options, often at the same time. A single issue at any level of the optimization can compromise the entire learning process. For example, in PPOC Klissarov et al. (2017), the number of different parametric functions to estimate grows linearly with the number of options, due to the need to estimate the policy, termination function and value for each option. This linear scaling increases significantly the likelihood of performance degradation, primarily due to destructive interference among the various estimators, which are typically represented withing a single neural netowrk with multiple heads. Consequently, this limits designers to using a restricted number of options, in order to avoid such interference.

To address the instability problem, in this paper, we take inspiration from boosting Freund et al. (1996), an approach for training complex non-linear classifiers which works by sequentially adding very simple, linear classifiers, are trained to solve a new version of the problem that is not solvable by their predecessors. Training proceeds until adding new classifiers does not provide any progress. While the mechanics of AdaBoost, the most famous boosting algorithm, are well suited for supervised learning, we use its idea as inspiration for an HRL approach which also adds options sequentially. Our approach limits the number of parameters which are allowed to change at any given time. If we want to learn $N$ options, this approach replaces a single, complex optimization of $O(3N)$ functions with a sequence of $N$ optimization problems, each of which optimizes 3 parameter sets. Each new optimization problem focuses only on the parameters of a single, new option, and on the

value function over options (which can be viewed as analogous to the weighting over hypotheses in boosting). The previous options are fixed, providing a trade-off between the quality of the solution obtained and the stability of the training process. Furthermore, the new abstractions build on top of previous knowledge: previously learned options guide exploration, facilitating the discovery of more complex behaviours. This is also analogous to the way in which in boosting, the already constructed weak classifiers shape the data distribution for the new classifiers being learned.

This paper contributes:

1. Option Boosting: An approach for controlling the stability of HRL optimization by constraining the learning to one option at a time.

2. Option Library: A method for knowledge sharing between agents, allowing concurrent agents to share option-policies.

We instantiate our ideas and evaluate them in two pixel-based, 2D gridworld environments, MiniGrid and CraftingWorld. Our approach enables stable option learning in both environments without requiring many tweaks, increasing the hope of developing empirically successful HRL methods.

## 2 BACKGROUND

In Reinforcement Learning (RL), an agent interacts with an environment, typically modelled as a Markov Decision Process (MDP) characterized by a tuple $M = (S, A, \gamma, r, P, s_0)$, where $S$ is the set of states and $A$ is the set of actions, the discount factor $\gamma \in [0, 1)$ de-values future rewards, $r : S \times A \to \text{Dist}(R)$ is the reward function, and $P : S \times A \to \text{Dist}(S)$ is the state transition function, defining the dynamics of the environment. Given a policy $\pi : S \to Dist(A)$ its value function, denoted $V_\pi(s)$, represents the expected return when starting from state $s$ and following policy $\pi$. The primary objective in RL is to find a policy $\pi$ that maximizes the value function at states sampled from the initial state distribution $s_0$. The value function is defined as: $V^\pi(s) = \mathbb{E}^\pi \left[ \sum_{t=0}^\infty \gamma^t r(S_t, A_t) \mid S_0 = s \right]$, refer to Sutton and Barto (2018) for an in-depth introduction to RL.

In HRL, the options framework provides a mechanism for learning and planning at both high- and low-level. An option set $\Omega = \langle \omega_0, \dots, \omega_k \rangle$ consists of a options $\omega$ defined by a tuple $\langle \pi, \beta \rangle$. Here, $\pi$ denotes the option-policy responsible for taking actions during the option's execution, and $\beta : S \to [0, 1]$ is the termination condition. The value of $\beta(s)$ represents the probability of the option terminating when the agent is in state $s$, with 0 meaning certain continuation and 1 indicating certain termination.

We assume that all options are available at all times in this work, thereby omitting the initiation set[1]. The policy-over-options, $\mu : S \to Dist(\Omega)$, takes control whenever an option terminates, selecting a new $\omega$ to execute according to the distribution $\mu(s)$. Once an option $\omega$ is activated by $\mu$, control is ceded to the option-policy $\pi$ until the termination condition $\beta$ is met. We denote by $\beta_k : S \to [0, 1]$ the termination condition for option $\omega_k$, representing the probability that $\omega_k$ terminates when the environment is in state $s'$.

We utilize the option set function $V_{\mu,\Omega}(s)$ which quantifies the value of executing $\mu$ and $\Omega$ in state $s$. The option-value functions are formally defined as:

$$V_{\mu,\Omega}(s) = \sum_k \mu(k|s) Q_\Omega(s, k),$$

$$Q_\Omega(s, k) = \sum_a \pi(a|s, k) Q_U(s, k, a),$$

where $Q_U : S \times \Omega \times A \to \mathbb{R}$ is the utility of taking a primitive action $a$ while committed to option $\omega$ in state $s$:

$$Q_U(s, k, a) = r(s, a) + \gamma \sum_{s'} P(s'|s, a) \left[ (1 - \beta_k(s')) Q^{\mu,\Omega}(s', k) + \beta_k(s') \max_{k'} Q^{\mu,\Omega}(s', k') \right].$$

---

[1]In the original options framework, an initiation set limits the states where an option can be initiated.

## 3   Method

The proposed algorithm seeks to discover "weak options" sequentially. The agent will then iteratively combine these optioons and build increasingly effective policies. Weak policies are designed to facilitate knowledge transfer across multiple tasks, while ensuring stability and improvement across iterations.

---

**Algorithm 1** Option Boosting

---

1: Define a space of weak policies as $\underline{\Pi} \subset \Pi$
2: Optimize for the first weak policy $\pi_0$:

$$\pi_0^* = \arg\max_{\pi \in \underline{\Pi}} V_\pi(\mathcal{S})$$

3: Define the initial optimization space $\Omega_0 = \langle \pi_0, \beta_0 \rangle$
4: $k \leftarrow 1$
5: **while** $\Delta V \geq \epsilon$ **do**
6:     Introduce new parameters $\mu$, $\beta_k$, $\pi_k$.
7:     Define the joint optimization space $\Omega_k = \Omega_{k-1} \cup \langle \pi_k, \beta_k \rangle$
8:     Optimize $\mu$, $\pi_k$, $\beta_{0:k}$ until a plateau 3.2.3 or the training budget runs out

$$(\mu^*, \pi_k^*, \beta_{0:k}^*) = \arg\max_{\mu, \beta_{0:k}, \pi_k \in \underline{\Pi}} V_{\mu, \Omega_k}(s_0)$$

9:     Compute improvement $\Delta V = V_{\mu, \Omega_k}(s_0) - V_{\mu, \Omega_{k-1}}(s_0)$
10:     $k \leftarrow k + 1$
11: **end while**
12: **Return** $\Omega_k$

---

### 3.1   Weak Policies

We constrain option policies to a space of "weak policies" as a form of regularization; this is to prevent over-specialization and facilitate the transfer of knowledge across tasks. To achieve this, in the multi-task setting, each option-policy $\pi^k$ is limited to using only information which does not reveal directly the identity of the task, and additionally, the capacity of the policy is constrained. For example, in our experiments in 2D environments, **each option-policy $\pi^k$ is limited to using only the pixel information** from the agent's field of view, without access to the task index. and uses neural networks of small capacity.

Commonly, we define policies by using parametric classes:

$$\Pi = \left\{ \pi_\theta \mid \theta \in \mathbb{R}^d \right\},$$

where $\Pi$ may not contain all stochastic policies (and it may not even contain an optimal policy). In the case of weak policies, $\pi^k \in \underline{\Pi} \subset \Pi$, where $\underline{\Pi}$ is a space even more constrained and less likely to contain an optimal policy. See section E.2, for a domain-specific definition.

On the other hand, the policy-over-options, value functions, and termination functions are allowed access to the full state information. This enables the higher-level components of the learning system to make informed decisions about which options to deploy and when to terminate them based on the current task and state. This provides a compromise between learning sub-behaviours that transfer while having a good overall policy.

### 3.2   Option Boosting

In the algorithm, $V_{\mu, \Omega_k}(s_0)$ is the value function at $s_0$ under the latest policy set $\Omega_k$ (including the learnable policy $\pi_k$), and $V_{\mu, \Omega_{k-1}}(s_0)$ is the value function at the initial state distribution $s_0$ under the previous policy set $\Omega_{k-1}$

### 3.2.1 FIRST STEP

The algorithmic step 2 is the first RL problem. For a weak policy space and an MDP, we want to find the optimal policy $\pi^* \in \underline{\Pi}$ that maximizes the expected discounted return. In the case of the first weak policy, there is no involvement of HRL methods, as we are optimizing for $V_\pi$, which represents the value function for a given weak policy $\pi$ over the original flat MDP.

### 3.2.2 SUBSEQUENT STEPS

In the subsequent steps of incremental option learning, the problem becomes a hierarchical reinforcement learning problem. We introduce a new weak policy, $\pi_k$, and a corresponding option termination function, $\beta_k$, to signal when the execution of $\pi_k$ terminates.

A high-level, $\mu$, provides the probability of picking each option in each state. We aim to optimize the high-level policy, the new weak policy $\pi_k$ and all the termination functions, to maximize the overall value function (step 8).

Finally, given a small value $\epsilon > 0$, the algorithm terminates when the improvement of the latest option-policy, $\pi_k$, is less than $\epsilon$:

$$V_{\mu,\Omega_k}(s_0) - V_{\mu,\Omega_{k-1}}(s_0) < \epsilon \tag{1}$$

### 3.2.3 STOPPING CRITERION

In the function approximation setting, the algorithmic step 8 necessitates an iteration budget. Due to the varying complexity of different tasks, a fixed threshold is unlikely to generate monotone improvement (as described in step 9 of the algorithm). To address this issue, we adopt an adaptive stopping criterion called BoostOnPlateau, which identifies improvements followed by plateaus in the learning process. This approach is inspired by learning rate reduction methods employed in supervised learning, such as ReduceLROnPlateau in PyTorch. [2]

$$\text{BoostOnPlateau}(m, p, r, b) = \begin{cases} \text{True}, & \text{if } (m - m_0) \geq r \text{ for } p \text{ consequent iterations} \\ \text{False}, & \text{otherwise} \end{cases} \tag{2}$$

The BoostOnPlateau function schedule consists of four parameters that collectively help to adjust the stopping criterion based on the observed progress in learning:

- **Performance metric** ($m$): This is the metric being monitored, which for RL is the estimated return. At the beginning of a stage, the performance metric has an initial value denoted $m_0$. As the learning process progresses, this metric is used to assess and adapt the stopping criterion based on the observed learning progress.

- **Reference metric value** ($m_0$): To reduce noise and have a more robust indicator of a learning plateau, we estimate $m_0$ by first learning a policy-over-options using only the frozen option policies from previous stages (**warm-up period**). Then, we introduce the new option-policy for a small number of steps (**value function stabilization period**); this is necessary to avoid spikes in the value estimation. $m_0$ is then set to the value estimate of the first policy rollout following these two phases.

- **Patience** ($p$): Patience determines the number of consecutive iterations without significant improvement before the stopping criterion is triggered. This allows the learning process to continue for a specified number of iterations in hopes of overcoming plateaus and achieving further improvement.

- **Required improvement** ($r$): The threshold parameter sets the minimum amount of improvement required to consider the progress significant. This helps to differentiate between actual learning progress and minor fluctuations in the metric.

The **performance metric** in our experiments is a smoothed estimated (by the agent) return [3].

$$\alpha m_{t-1} + (1 - \alpha)\hat{V}_t^{\mu,\Pi} \tag{3}$$

---

[2] https://pytorch.org/docs/stable/generated/torch.optim.lr_scheduler.ReduceLROnPlateau.html

[3] In the tasks considered for this paper, many of which involve sparse reward settings, relying solely on unbiased metrics such as the rollout reward tends to result in higher variance. This introduces challenges in

### 3.3 Option library

So far, we have been considering a single task setting, but the sequentiality of our proposed boosting method provides the ability to transfer knowledge effectively also between different agents. To do so, we introduce the Option Library, a mechanism that allows concurrent agents to share frozen options at each boosting stage. This enables knowledge transfer not only across tasks but also across different agents working on similar or related tasks.

An option library can be thought of as a repository of option-policies that have been learned by other agents during their training. As a new learning phase starts (either after a boosting event or after a long time of no improvement), the options stored in the library are read and transferred to the agent which can use them as if they were its own.

The benefits of using an Option Library include:

- **Transfer Learning:** Reusing previously learned option-policies can significantly reduce the learning time for new tasks, as agents can leverage other agents' knowledge.
- **Scalability:** The Option Library facilitates multi-agent collaboration by allowing agents to learn in parallel while contributing to and benefiting from a shared pool of knowledge. This concurrent learning and knowledge transfer mechanism substantially enhances the system's ability to scale across numerous tasks and agents.
- **Ease of Design:** The Option Library not only enables the designer to implicitly define option-policies through the MDP and let the parameterised form of the option-policy to be optimized, but also provides designers with valuable insights into their utility. By examining the frequency with which an option is invoked across agents, a designer can gauge its effectiveness and applicability across tasks. This serves as a practical metric for evaluating the contribution of each task by judging its options within the library.

To illustrate the utility of the Option Library, consider a scenario involving multiple agents learning various tasks in an environment with locked doors and keys. One agent may be tasked with learning to pick up keys, while another agent focuses on opening locked doors. By sharing the learned option-policy related to key manipulation through the Option Library, the door-opening agent can benefit from the knowledge acquired by the key-picking agent, allowing it to learn how to approach and use keys to unlock doors more efficiently.

## 4 Experimental setup

To highlight our proposed incremental Option Boosting approach and Option Library, we designed an experimental setup that includes diverse environments of increasing complexity focusing on transferability across environments.

### 4.1 Environments

Figure 1 shows a visual representation of the environments used to measure the effectiveness of this method.

#### 4.1.1 Minigrid

Minigrid offers a variety of first-person gridworld tasks designed for evaluating RLalgorithms. The agent has a first-person view of the area in front (see highlighted area) and can move forward, turn, or pick up items.

**Distractors** To increase the complexity of the minigrid tasks, we introduce distractors in the form of task foreign items. If the agent interacts with these items, it will receive a negative reward and terminate the task. These distractors challenge the agent's ability to focus on the primary task and encourage the development of more robust and transferable option-policies.

---

reliably determining learning plateaus. To mitigate this issue, we opt for the smoothed estimated return as our performance metric. The estimated return, is generally more stable and less prone to fluctuations in these tasks, making it an empirically robust choice for monitoring the learning progress.

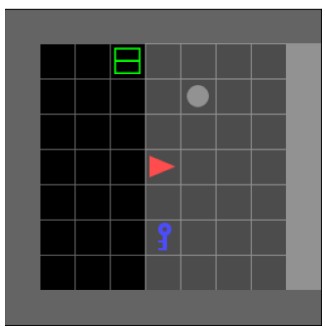

(a) PickupXLoc: The agent must pick up an X (X could be ball, lock, box).

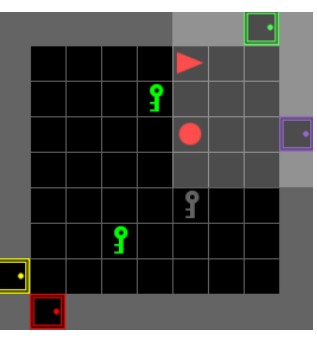

(b) PickupOpenLoc: The agent must pick up an object or open a door depending on the mission instruction.



(c) OpenLoc: The agent must pick up an object or open a door depending on the mission instruction.

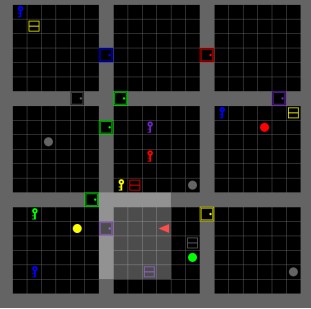

(d) Open: The agent must navigate a 9-room environment and open a door of the correct colour.

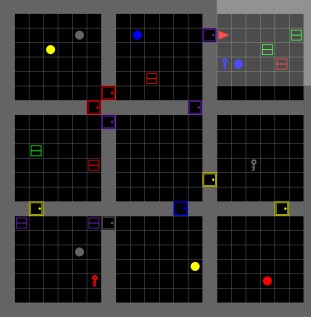

(e) Pickup: The agent must navigate a 9-room environment and open a door of the correct colour.

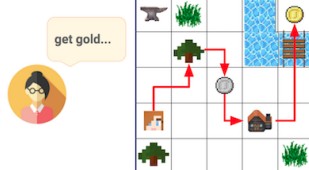

(f) Crafting environment, all environments are the same, what changes is the task. In the example, the agent needs to first collect wood and transform it at a workshop to construct a bridge, to finally pass across the water and reach the gold.

Figure 1: Visualization of RL environments, 1a to 1e are visualizations of the minigrid environments while 1f is an interpretation of the crafting environment

### 4.1.2 CRAFTINGWORLD

By incorporating CraftingWorld into our experimental setup, we aim to show the generality of the method across different environment classes.

CraftingWorld is a highly flexible 2D environment adapted from Andreas et al. (2017) and further extended to support a myriad of hierarchical tasks in a procedurally generated setting [4]. The environment involves a gamut of tasks that require both navigational and crafting abilities. Figure 1f offers an example where the agent needs to execute a sequence of steps to complete a "pick up gold" task.

In CraftingWorld, the agent is capable of moving in the 2D grid space (up/down/left/right), picking up objects to store them in an inventory, and utilizing workshops to transform collected items. For example, in the task "pick up gold," the agent has to perform the following sequence of high-level actions: Collect wood. Utilize a workbench to transform wood into a plank. Acquire iron and use a factory to make a bridge. Use the crafted bridge to cross water and finally collect gold. The environment houses a total of 17 tasks of varying complexity, from simple item collection ("Get X") to complex crafting sequences requiring multiple steps and sub-tasks. These tasks provide a thorough evaluation of our method's ability to handle sparse rewards and extended action sequences.

---

[4]see: https://github.com/Feryal/craft-env

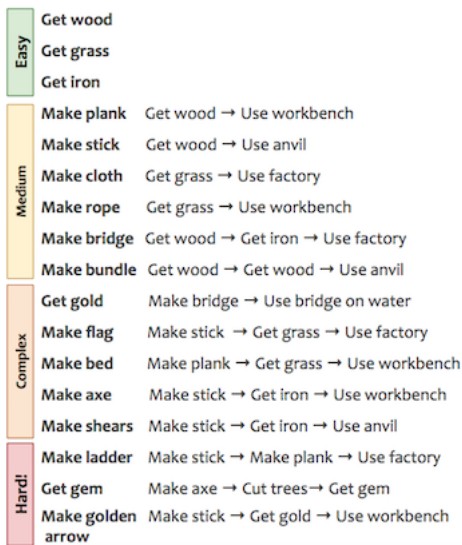

Figure 2: Comprehensive list of tasks available in CraftingWorld, ranked by complexity.

## 4.2 EVALUATION METRICS

We evaluate the performance of our proposed method based on two main metrics: the average performance across all tasks and the number of successful tasks completed. The average performance represents the optimized objective function, while the number of solved tasks is an indicator of generality across tasks.

A task is deemed solved when it attains positive rewards in over 20% of the rollouts. While this metric may appear easily achievable, it presents a significant challenge due to the hard exploration nature of all the tasks involved.

## 4.3 BASELINE METHODS

The primary aim of our study is to investigate the efficacy of our sequential boosting method, rather than a broad comparison with state-of-the-art algorithms. While we employ PPOC as a baseline estimator, it is important to note that our method is agnostic to the choice of baseline.

1. PPO: Schulman et al. (2017) The agent is trained independently on each task, without any knowledge transfer between tasks, we use this agent as a baseline.

2. PPOC : A hierarchical reinforcement learning method that uses options for learning policies, to highlight the effect of boosting and option-policy sharing.

## 4.4 SUCCESSFUL TASKS

| Method | Minigrid | Craft Env |
|---|---|---|
| Single Task Baseline | 6/7 | 13/17 |
| PPOC | 6/7 | 13/17 |
| Boosting | 7/7 | 14/17 |

## 4.5 Average (4 seed) task return

| Task name | Per Task PPO | Option Boosting | PPOC(4) | Improvement vs PPO | Improvement vs PPOC(4) |
|---|---|---|---|---|---|
| PickupOpenLoc | 0.115 | 0.959 | 0.335 | 0.844 | 0.624 |
| Pickup | -0.980 | -0.336 | -1.000 | 0.645 | 0.664 |
| PickupKeyLoc | 0.963 | 1.000 | 1.000 | 0.037 | -0.000 |
| PickupBoxLoc | 0.970 | 1.000 | 1.000 | 0.030 | -0.000 |
| OpenMidLoc | 1.000 | 1.000 | 0.993 | 0.000 | 0.007 |
| Open | 1.000 | 1.000 | 0.857 | 0.000 | 0.143 |
| PickupBallLoc | 1.000 | 0.997 | 1.000 | -0.003 | -0.003 |
| minigrid mean | 0.508 | 0.803 | 0.648 | 0.294 | 0.155 |
| make[ladder] | 0.000 | 1.000 | 0.000 | 1.00 | 1.000 |
| make[bridge] | 0.983 | 0.994 | 0.995 | 0.004 | -0.001 |
| make[rope] | 0.996 | 1.000 | 0.999 | 0.003 | 0.001 |
| make[cloth] | 0.990 | 0.994 | 0.993 | 0.003 | 0.001 |
| make[shears] | 0.993 | 0.996 | 0.992 | 0.000 | 0.004 |
| make[hammer] | 0.683 | 0.683 | 0.697 | 0.000 | -0.014 |
| make[flag] | 0.000 | 0.000 | 0.000 | 0.000 | 0.000 |
| make[goldarrow] | 0.000 | 0.000 | 0.000 | 0.000 | 0.000 |
| get[gem] | 0.000 | 0.000 | 0.000 | -0.001 | 0.000 |
| make[knife] | 0.992 | 0.991 | 0.992 | -0.001 | -0.001 |
| get[grass] | 1.000 | 0.999 | 1.000 | -0.046 | -0.001 |
| get[wood] | 1.000 | 0.954 | 1.000 | -0.047 | -0.046 |
| make[plank] | 1.000 | 0.953 | 0.994 | -0.101 | -0.041 |
| make[stick] | 0.997 | 0.896 | 0.996 | -0.211 | -0.100 |
| make[bundle] | 0.689 | 0.479 | 0.694 | -0.215 | -0.215 |
| get[iron] | 1.000 | 0.785 | 1.000 | -0.247 | -0.215 |
| get[gold] | 0.703 | 0.456 | 0.952 | -0.250 | -0.496 |
| make[bed] | 0.997 | 0.747 | 0.964 | -0.300 | -0.217 |
| make[axe] | 1.000 | 0.700 | 1.000 | -0.300 | -0.300 |
| crafting mean | 0.626 | 0.693 | 0.751 | 0.012 | -0.058 |

### 4.5.1 Results and Discussion

When comparing with PPO-OptionCritic, we observe that our method shows a more efficient and flexible learning process. As the number of options in PPO-OptionCritic increases, the overall performance improves but at the cost of increased complexity and computational requirements. Our incremental option learning approach, on the other hand, adapts the number of options based on performance improvement, ensuring a more efficient learning process.

By carefully examining the experimental results, we aim to demonstrate the effectiveness of our proposed incremental option learning approach and Option Library in multi-task learning and knowledge transfer. Furthermore, we highlight the advantages of our method in terms of learning efficiency, scalability, and collaboration between agents in complex environments.

Our empirical findings underscore the efficacy of the proposed boosting approach in tandem with the Option Library, allowing the agent to solve harder scenarios. While the method trades off optimality for knowledge transfer across tasks, it opens avenues for further optimization.

In comparison with PPOC equipped with four options, it becomes evident that augmenting the option set up to about 15 options through the Option Library negatively impacts both performance and stability; for instance, the 'make[gold]' task was completed in only half of the runs.

While further refinement in policy determinism and hyperparameter tuning could yield improvements, we consciously maintained an equal footing in hyperparameter optimization between the methods, in both computational and human-effort terms.

## 5 RELATED WORK

The ability to bootstrap from simpler policies to more complex policies has been the focus of many HRL methods. A common approach is to define a guiding function, such as an intrinsic reward or a curiosity mechanism, to guide the emergence of the hierarchical structure Barto et al. (2004).

The Options framework has lead to recent approaches Levy and Shimkin (2011); Bacon and Precup (2017); Harb et al. (2018a); Riemer et al. (2018); Tiwari and Thomas (2019) that rely on policy gradient methods to learn the option policies, these approaches make no effort of controlling the balance between learning and forgetting, making the learning process unstable.

In supervised learning, the idea of freezing old parameter sets has been explored in progressive nets Rusu et al. (2017) to solve complex task sequences while leveraging transfer and avoiding catastrophic forgetting. Similar to our method, their approach is immune to forgetting and can leverage prior knowledge through lateral connections to previously learned features. However, unlike our approach, once a task has been learned, no new knowledge can be extracted from it, while in our case, a frozen policy is still useful to guide exploration.

Our work differs from these approaches in that we control option-instability by limiting parameter changes at any given time. This method promotes the discovery of reusable abstractions, allowing learned options to guide exploration and facilitate the discovery of more complex behaviours. We verify our approach through experiments and results that demonstrate the effectiveness of our method in stabilizing learning within the deep reinforcement learning context.

## 6 CONCLUSION AND FUTURE WORK

We introduced an approach apt to multi-task learning in reinforcement environments, focusing on incremental option learning and a shared Option Library. The primary contributions of this paper lie in three key areas: transfer learning, scalability, and multi-agent collaboration.

We demonstrated that our incremental option boosting approach consistently transfers knowlege across tasks in hard environments of the minigrid suite augmented with distractors and solves 14/17 tasks in the Crafting environment, compared to both the single task and PPOC baselines which solve 13/17.

The Option Library not only reduced the learning time but also scaled effectively across multiple agents. In contrast to other tabula-rasa algorithm, it is possible to "seed" a library with previously discovered option-policies allowing for easier transfer.

However, the study is not without its limitations. While our approach provides a suggestion to how to generate knowlege, future work could investigate the elimination strategies for options in the library as agents might contribute similar options or a later agent might learn a better version of the same skill, a recurring theme across the experiments is the negative effect of having too many, often redundant option-policies.

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

## A    TRAINING DETAILS

When training agents with PPOC a lot of intuition is shared with Proximal Policy Optimization (PPO) with an Adam optimizer, as it's the underlying estimator used for the various modules. To train, we run multiple training sessions, each comprising a fixed number of samples from the environment. Each episode's length is limited to prevent excessively long training times.

We train each agent with the proposed incremental option learning approach and the Option Library, each agent is learning about different tasks in similar environments.

### A.1    META-POLICY NETWORK STRUCTURE

The network structure of our meta-policy and the option-policies are kept invariant across experiments except for the features extractor layer which is specialized for the minigrid tasks.

The meta-policy network, `InitializationMetaActorCriticPolicy`, consists of the following components:

1. **Features Extractor**: A TaskEmbedding1CNNFeaturesExtractor module, which includes:
   - An image processing pipeline composed of AliasBlacks (maps anything that renders as black to the same value; only used for minigrid), ObjectDetector (embeds different object ids to a random higher dimensional vector; only minigrid), and Simple1x1CNN layers.
   - Two linear layers for mission colour and mission object embeddings.
2. **MLP Extractor**: An MlpExtractor module that processes the features obtained from the Features Extractor. This module contains separate policy and value networks.
3. **Action Network**: A linear layer that outputs action probabilities of different options.
4. **Value Network**: A linear layer that estimates the value function.
5. **Initialization**: An Initiation module, which includes:
   - A FeaturesExtractor for feature extraction.
   - Option termination modules, each containing a linear layer followed by a sigmoid activation function.
   - Option initialization is controlled by the boosting process, every time we "add an option" we are actually marking that option as available anywhere by changing the constant initialization probability from 0 to 1.

### A.2    OPTION POLICY NETWORK STRUCTURE

The option policy network, `SimpleACNoTask`, has the following components:

1. **Features Extractor**: A TaskEmbedding1CNNFeaturesExtractor module, which includes:
   - An image processing pipeline composed of AliasBlacks (see above), ObjectDetector (see above), and Simple1x1CNN layers.
   - Two linear layers for mission colour and mission object embeddings.
2. **MLP Extractor**: A ShatterFeatures module containing a MlpExtractor, which includes separate policy and value networks but hides task information for the option-policy network.
3. **Action Network**: A linear layer that outputs action probabilities for different actions.
4. **Value Network**: A linear layer that estimates the value function.

## B    HYPERPARAMETER EXPLORATION

In this appendix, we provide an in-depth exploration of the hyperparameters used in our experiments, such as the metric smoothing factor, patience, threshold, and smoothing weight. By investigating their impact on the performance of our proposed method, we aim to strengthen the paper's conclusions and offer more detailed information to reuse the method.

### B.1 METRIC SMOOTHING FACTOR

The metric smoothing factor controls the degree of smoothing applied to the performance metrics when evaluating the agent's progress during training. A higher smoothing factor reduces the noise in the performance metrics, making it easier to identify trends and plateaus in the agent's learning process. However, if the smoothing factor is too high, it may obscure short-term improvements and slow down the agent's learning. The smoothing factor was empirically set by disabling boosting (setting patience to infinity) and observing when the performance improvement would plateau on a subset of simpler tasks, then set conservatively.

### B.2 PATIENCE

Patience determines the number of iterations the algorithm waits before considering a plateau in performance. If the agent's performance does not improve for a specified number of iterations, the algorithm may decide to add a new option-policy or terminate the training process. A higher patience value allows the agent more time to explore and exploit the current option-policy set, potentially leading to more stable learning. On the other hand, a lower patience value may result in more frequent updates to the option-policy set, which could increase the risk of over-fitting or premature convergence. We performed a grid search to identify the most suitable patience value for our experiments, ensuring that the agent has adequate time to learn while still responding promptly to performance plateaus.

The patience parameter was set conservatively as generating too many sub-optimal option-policies increases the number of options and actions and exploration complexity of the HMDP.

### B.3 THRESHOLD

The threshold parameter controls the minimum improvement required to consider an update to the option-policy set as beneficial. A higher threshold encourages the algorithm to focus on more substantial improvements and prevents minor fluctuations in performance from triggering unnecessary updates. However, if the threshold is too high, the agent may miss valuable opportunities to learn from smaller improvements. We experimented with various threshold values and selected the one that achieved the best trade-off between stability and sensitivity to performance changes.

The initial guess of 0.05 worked robustly and was never changed.

### B.4 HYPERPARAMETER SWEEP

In the course of our experiments, we used a structured approach to select the optimal hyperparameters for our model, we made extensive use of hyperparameter sweeps. The details of our sweep are as follows:

The hyperparameter sweep was performed by executing the algorithm with varying parameters and recording the resulting metrics. The metric of interest was the `smoothed_mean_returns`, which we aimed to maximize. The method used for the hyperparameter optimization was Bayesian optimization.

The hyperparameters of interest were as follows:

- `entropy_regularization_initial`, `lr_mu`, and `lr_pi` followed a log-uniform distribution within the range $[10^{-4}, 10^{-1}]$.
- `rollout_steps` and `num_envs` followed a quantized log-uniform distribution with respective ranges [4, 256] and [4, 512].
- `batch_size_fraction` was set to 1.
- `n_epochs` could be either 1 or 2.
- `normalize_advantage` could be either False or True.
- `object_embedding_size`, `option_cnn_channels`, `option_features_dim` had possible values of 32, 64, or 128.
- `option_heads_size` could be 32, 64, 128, or 256.

- `meta_policy_heads_size` could be 32, 64, 128, or 256.
- `meta_policy_cnn_channels` could be 8, 16, or 32.
- `meta_policy_features_dim` could be 32, 64, or 128.

In our sweep, each hyper-parameter configuration was tested and the performance was evaluated based on the aforementioned metric. The configuration that resulted in the maximum value of `smoothed_mean_returns` was selected as the optimal set of hyperparameters for our model. Both baseline and method benefited to approximately the same amount of hyper-parameter optimization wallclock time.

## C  PERFORMANCE ANALYSIS

In this section, we delve deeper into the performance analysis of our proposed method across different tasks. Our goal is to identify the types of tasks where our method excels or struggles, offering valuable insights into potential areas for future improvement or limitations of the approach.

### C.1  TASKS WHERE THE METHOD EXCELS

Our method showed exceptional performance in tasks that involve a combination of navigation, object manipulation, and interaction with various items. The incremental option learning approach and the Option Library effectively transfer knowledge between these tasks, allowing the agent to leverage prior knowledge and avoid learning from scratch. As a result, the agent demonstrates superior performance in tasks where the skill sets required overlap considerably, but the exploration difficulty is complex.

Another area where our method excels is in environments with distractors. The agent learns to focus on the primary task while ignoring irrelevant items, improving the overall robustness and transferability of the option-policies. This ability to handle distractors effectively demonstrates the adaptability and resilience of our proposed method in complex environments.

### C.2  TASKS WHERE THE METHOD STRUGGLES

While our method performs OK in most tasks, converging to optimal policies is a challenge due to the regularization effect imposed by design and the optimality price to pay when exploring a bigger number of options. Moreover, the knowledge transfer through the Option Library might be less effective, as the previously learned option-policies may no longer be relevant or applicable, and even if applicable, since we are not improving them, they might not generalize as well to similar new situations.

Another potential limitation is in tasks where the optimal policy is highly specialized and does not overlap with previously learned tasks. In these cases, the incremental option learning approach and the Option Library may not provide significant benefits, as the agent has to learn the new policy from scratch, our method actually hinders optimization as the agent now has destructive interference from previous skills and has to learn to ignore them. However, this limitation is expected, as our method primarily focuses on leveraging knowledge transfer between tasks with similar skill sets or overlapping goals.

## D  COMPUTATIONAL REQUIREMENTS

In this section, we provide a detailed discussion of the computational requirements of our proposed method, including its computational complexity, memory requirements, and training time. Understanding these aspects is crucial for evaluating the practicality of our approach in real-world applications.

Running the experiment for the 17 environments in crafting world requires 2 days of wall-clock time (simpler tasks can be stopped earlier), and 14 compute days overall. For the 7 mini-grid experiments we require 3 wall clock days (first-person rendering and occlusions slow down the environment), and

use 8.5 compute days overall. Most of the computing was done on the ComputeCanada cluster, with 16GB memory nodes.

# E    MINIGRID

## E.1    OBSERVATION SPACE

The state space $\mathcal{S}$ of the Markov Decision Process (MDP) is defined as the cross product of the pixel state space $\mathcal{S}_{\text{pixels}}$ and the mission description state space $\mathcal{S}_{\text{mission}}$ [5]:

$$\mathcal{S} = \mathcal{S}_{\text{pixels}} \times \mathcal{S}_{\text{mission}} \tag{4}$$

## E.2    WEAK POLICIES

A *weak policy* is defined as a policy $\pi$ that is limited to using only the pixel information from the agent's field of view, without access to the task index.

# F    REMARKS:

## F.1    POLICY FREEZING IN ALGORITHM

When we refer to a policy as "frozen," we imply that its parameters remain fixed during subsequent learning phases. Specifically, when learning $\pi_k$, the parameters of all previous policies $\pi_0, \ldots, \pi_{k-1}$ are kept constant.

## F.2    ON GUARANTEED IMPROVEMENT

We aim to demonstrate that the optimal value function $V_{\mu,\Omega_k}(s)$ is no less than $V_{\mu,\Omega_{k-1}}(s)$ for all $s \in S$, when adding a new option to the set.

**Proof:**    Consider an optimal policy $\mu_{\text{orig}}$ for the original option set $\Omega_{k-1}$. We can construct a new policy $\mu_{\text{exp}}$ such that for each state $s$,

$$\mu_{\text{exp}}(s) = \mu_{\text{orig}}(s).$$

Therefore, we have $V_{\mu,\Omega_k}(s) \geq V_{\mu,\Omega_{k-1}}(s)$, proving that the expanded option set does not degrade performance.

## F.3    ENTROPY SCALING

To maintain a balance on exploration when adding a new option, we adjust the entropy coefficient at every round by a factor $s$ of:

$$s = \frac{\log(n_2)}{\log(n_1)}.$$

This is based on the entropy $H(\pi)$ for a uniform distribution over $n$ actions is $\log(n)$. To equate the entropies for two uniform distributions over $n_1$ and $n_2$ actions. However, we found that the specific scaling factor was not critical; a linear schedule performed similarly.

---

[5]we are ignoring the agent direction, which is not necessary for these tasks.

