# OpenReview forum: "Option Boosting"
_ICLR.cc/2024/Conference — Submitted to ICLR 2024_

### Official Review · Reviewer_Sp3D · 2023-10-12

**Soundness:** 2 fair
**Presentation:** 1 poor
**Contribution:** 2 fair
**Rating:** 3
**Confidence:** 4

**Summary:**

This paper advocates for the sequential learning of options to bolster the stability of Hierarchical Reinforcement Learning, drawing inspiration from supervised learning methodologies. However, it falls short in terms of presentation quality and experimental validation. Notably, the proposed algorithm fails to demonstrate significant improvement over weak baselines, leading me to recommend rejection of this paper.

**Strengths:**

The algorithm design is well-conceived, aiming to enhance the stability of Hierarchical Reinforcement Learning (HRL) by employing a boosting technique derived from supervised learning. Additionally, the proposed HRL algorithm holds promise for multi-task reinforcement learning.

**Weaknesses:**

(a) For the related work part, the authors should put an emphasis on Hierarchical Reinforcement Learning and provide comparisons with them to show the novelty. Works on option discovery (a.k.a., skill discovery) should also be included.

(b) Sequentially learning options and restricting them as weak policies could potentially lower the sample efficiency and option usability.

(c) The criterion design in this algorithm is complicated and includes multiple hyperparameters, which may be hard to be fine-tuned.

(d) The paper needs quite some polish, such as the layout of Figure 1 and quality of Figure 2.

(e) The baselines are too weak to be supportive. More recent and advanced HRL algorithms should be included, given the extensive research in this area.

(f) The results in Section 5 cannot show the advance of the proposed algorithm.

**Questions:**

(a) $Q^{\mu, \Omega}$ is not defined in Section 2.

(b) The definition of the weak policy space should better be presented in the main context. Also, its design is domain-specific, which may deter the generality of the proposed algorithm.

(c) The algorithm design involves multiple key hyperparameters shown in Eq. (1)-(3). Sensitivity analysis regarding them can be beneficial to show the robustness of the proposed algorithm.

(d) Fixing options learned from early training stages can potentially make them under-performed and less prefered and thus never be used in the late training stages.

---

### Official Review · Reviewer_Jjah · 2023-11-01

**Soundness:** 3 good
**Presentation:** 2 fair
**Contribution:** 2 fair
**Rating:** 3
**Confidence:** 4

**Summary:**

The paper applies the idea of boosting from ensemble learning within the options framework of Hierarchical Reinforcement Learning (HRL), progressively generating new options while fixing and reusing previously trained options. Additionally, the authors introduce an option library to facilitate knowledge transfer between different tasks and agents during multi-task training. Experiments were conducted to validate this approach in two types of environments.

**Strengths:**

The idea presented in the article is intriguing, attempting to integrate the boosting method from ensemble learning to reduce the difficulty of learning individual options and strategies, while also allowing for the reuse of prior knowledge in subsequent learning. The proposed approach is quite intuitive and straightforward, making it easy to follow.

**Weaknesses:**

The main issue is that the experimental results are somewhat weak, with only a few and relatively weak comparison baselines. The improvement from the Boosting method introduced in the paper is also limited, with both comparison baselines, PPOC and PPO, achieving the same number of task successes, and the paper's method only adding one success in each case. Moreover, in the Crafting environment, the paper's method underperforms PPO and PPOC in many tasks, and the overall performance differences are not significant. Another aspect is that the related work is not sufficiently comprehensive, with the latest work seemingly from 2019 (I am uncertain if there are no updates on HRL or Options framework since then).

Additionally, if different environments are considered as different tasks, the paper's setting seems to relate to curriculum design or automatic environment design and multi-agent policy learning and reuse, suggesting that these areas [1-3] may provide new insights for the work presented in this paper.



1. NeurIPS'21. Replay-Guided Adversarial Environment Design.
2. ICLR'23. Open-Ended Environment Design for Multi-Agent Reinforcement Learning.
3. ICLR'21. Iterative Empirical Game Solving via Single Policy Best Response.



Minor：some typos

(1) Section 3, optioons --> options

(2) Section 4.1.1 RLalgorithms --> RL algorithms

**Questions:**

1. How is the Option represented in this article? Is it a task index or embeddings output in the intermediate network? Also, does the Option in this paper have a practical meaning, or does it correspond to certain features/representations?
2. How is the V function in inequality (1) calculated?
3. Section 3.1 states "See section E.2, for a domain-specific definition." However, section E.2 contains only the same content  highlighted in bold as in the main text of Section 3.1. Does "weak policies" refer to those without access to the task index and using a smaller capacity network?
4. The experiments used 4 seeds, but the results do not report the variance.

---

### Official Review · Reviewer_ZULE · 2023-11-04

**Soundness:** 2 fair
**Presentation:** 3 good
**Contribution:** 2 fair
**Rating:** 5
**Confidence:** 3

**Summary:**

The paper introduces a novel methodology named "Option Boosting" for enhancing stability and knowledge transfer in multi-task hierarchical reinforcement learning (HRL). This method addresses the challenges faced by modern Hierarchical Reinforcement Learning (HRL) algorithms, particularly instability due to the multifaceted nature of the optimization process. By drawing inspiration from boosting methods, the paper proposes a mechanism wherein each option (a policy in HRL) is limited expressively to avoid abrupt knowledge changes. Through experiments, the paper demonstrates the capability of the proposed approach to effectively transfer knowledge from simpler tasks to complex ones in two pixel-based 2D gridworld environments: MiniGrid and CraftingWorld.

**Strengths:**

The paper offers a unique approach to address the instability in HRL, drawing inspiration from boosting techniques. The concept of controlling option-instability through limited expressiveness is original and significant. The paper also succeeds in demonstrating the transfer of knowledge from simpler to more complex tasks. The introduction of an incremental option boosting mechanism and the Option Library, which facilitates multi-agent collaboration, are noteworthy contributions.

**Weaknesses:**

The paper does not provide a comparative analysis with other state-of-the-art methods in HRL. While the paper showcases the effectiveness of the method in two specific gridworld environments, its applicability and scalability in more complex or diverse settings remain unexplored. There's also a concern about the potential negative effect of having redundant option-policies.

**Questions:**

1. How does Option Boosting compare to other existing methods in HRL in terms of efficiency and performance?
2. Can the proposed approach handle more complex environments beyond the gridworld scenarios?
3. How can the issue of redundant option-policies be effectively addressed in the proposed framework?

---

### Meta-Review · Area_Chair_UsbQ · 2023-12-14

**Metareview:**

The paper suggests an interesting approach to address an instability issue of HRL within options, based on a boosting method. While the approach is novel, there are several concerns from the reviewers, especially regarding evaluations -- weak baseline comparison (no comparison against SOTA in HRL) and relatively easy environments only. Also, there was no rebuttal.

**Justification For Why Not Higher Score:**

Weak evaluations and no rebuttal

**Justification For Why Not Lower Score:**

N/A

---

### Decision · Program_Chairs · 2024-01-16

Reject